# Identification of Elite Alleles and Candidate Genes for the Cotton Boll Opening Rate via a Genome-Wide Association Study

**DOI:** 10.3390/ijms26062697

**Published:** 2025-03-17

**Authors:** Qi Ma, Xueli Zhang, Jilian Li, Xinzhu Ning, Shouzhen Xu, Ping Liu, Xuefeng Guo, Wenmin Yuan, Bin Xie, Fuxiang Wang, Caixiang Wang, Junji Su, Hai Lin

**Affiliations:** 1College of Life Science and Technology, Gansu Agricultural University, Lanzhou 730070, China; qmacotton@163.com (Q.M.); 15117242582@163.com (X.Z.); guoxuefeng2025@163.com (X.G.); yuanwm2022@163.com (W.Y.); fxwang2023@163.com (F.W.); wangcaix@gsau.edu.cn (C.W.); 2Institute of Cotton Research, Xinjiang Academy of Agricultural and Reclamation Sciences, Shihezi 832000, China; lijilian@sohu.com (J.L.); ningxz@sohu.com (X.N.); xu.shouzhen@foxmail.com (S.X.); liu19901070@126.com (P.L.); xiebin1134@163.com (B.X.)

**Keywords:** upland cotton, natural population, boll opening rate, genome-wide association study, candidate genes

## Abstract

The boll opening rate (BOR) is an early maturity trait that plays a crucial role in cotton production in China, as BOR has a significant effect on defoliant spraying and picking time of unginned cotton, ultimately determining yield and fiber quality. Therefore, elucidating the genetic basis of BOR and identifying stably associated loci, elite alleles, and potential candidate genes can effectively accelerate the molecular breeding process. In this study, we utilized the mixed linear model (MLM) algorithm to perform a genome-wide association study (GWAS) based on 4,452,629 single-nucleotide polymorphisms (SNPs) obtained through whole-genome resequencing of a natural population of 418 upland cotton accessions and phenotypic BOR data acquired from five environments. A total of 18 SNP loci were identified on chromosome D11 that are stable and significantly associated with BOR in multiple environments. Moreover, a significant SNP peak (23.703–23.826 Mb) was identified, and a *GH-D11G2034* gene and favorable allelic variation (GG) related to BOR were found in this genomic region, significantly increasing cotton BOR. Evolutionary studies have shown that *GH-D11G2034* may have been subjected to artificial selection throughout the variety selection process. This study provides valuable insights and suggests that the *GH-D11G2034* gene and its favorable allelic variation (GG) could be potential targets for molecular breeding to improve BOR in upland cotton. However, further research is needed to validate the function of this gene and explore its potential applications in cotton breeding programs. Overall, this study contributes to the advancement of genetic improvement in early maturity and has important implications for the sustainable development of the cotton industry.

## 1. Introduction

Cotton (*Gossypium* spp.) is a vital cash crop that produces fiber, protein, and edible oil simultaneously [1,2] and is an important strategic reserve material associated with the national economy and people’s livelihood and security [3]. China’s cotton production was initially centered mainly in the Yangtze River region (YZRR) and Yellow River region (YRR) and gradually evolved into the “three pillars” of the YZRR, YRR, and Northwest Inland region (NIR) [4]. However, owing to the widespread competition for arable land for grain and cotton production, the NIR region rapidly grew into the largest and most important cotton production region in China.

In the NIR, the temperature decreases quickly in autumn, and the first frost appears early; in contrast, the temperature increases slowly in spring, and the final frost ends relatively late [5,6]. Therefore, early-maturing cotton varieties can reasonably be planted to make use of the limited frost-free period in this environment [7]. With the simultaneous severe shortage of labor resources and a sharp increase in the cost of manual cotton picking, mechanical harvesting has become the most widely used cotton harvesting method in the NIR [8,9]. The plants must be sprayed with defoliant before mechanical harvesting because cotton in this region cannot naturally mature and defoliate. However, spraying defoliants leads to a decrease in cotton yield and fiber quality [7]. To reduce the impact of chemical defoliants on cotton fibers, the overall boll opening rate (BOR) must exceed 50% [9,10,11]. However, in actual production, most varieties do not mature early, and most cotton in the NIR range sprayed with defoliation and ripening agents has a BOR under 50%, which severely affects the fiber yield and quality of mechanically harvested cotton. In actual production, this impact is manifested in two main aspects. First, a lower BOR inevitably leads to incomplete defoliation, increasing the impurity content of machine-picked cotton, increasing the frequency of impurity removal during cotton processing, and thereby reducing the length loss of cotton fibers by 1–2 mm. Second, a lower BOR can cause 1–2 cotton bolls at the top of the plant to fail to open naturally, and spraying defoliant can reduce the weight of a single cotton boll by approximately 0.5 g. Consequently, BOR is a highly important trait to consider when trying to improve the quality and yield of mechanically harvested cotton.

Although numerous indicators of early maturity in cotton have been identified, BOR is one of the most pivotal indicators [12]. The boll opening period refers to the date of the first boll opening of half of the cotton in the field [7]. Although some varieties have been shown to have a boll opening period that begins early, this period is long [13], which suggests that the boll opening period alone cannot completely reflect the early maturity of cotton varieties. However, a boll opening period that is too long affects the mechanized harvesting process. Therefore, BOR has become a key factor of early maturity and has played an increasingly prominent role in the production of machine-harvested cotton. Recently, with the widespread promotion of mechanized harvesting of cotton in China, BOR has received increasing attention [7,13]. However, BOR is a complex quantitative trait determined by quantitative trait loci (QTLs). Therefore, in-depth analysis of the genetic basis of BOR, identification of related QTL alleles, and exploration of candidate genes are highly important for the cultivation of new cotton varieties suitable for mechanical harvesting.

As early as 1988, association analysis was demonstrated to be more effective than the commonly used linkage analysis methods in identifying and elucidating the genetic mechanisms of complex diseases, and the concept of a genome-wide association study (GWAS) was proposed [14]. GWAS has gradually been widely recognized as one of the most effective methods for detecting crop genome-wide QTLs based on linkage disequilibrium (LD) [15,16]. Unlike two-parent cross-mapping, a GWAS uses many individuals with rich natural variation as research objects to analyze the role of different alleles at a given locus [17]. In addition, a GWAS based on LD uses natural populations for localization directly, omitting the population construction process, through which associated loci can be quickly located with high localization accuracy [18]. Additionally, only one genotype of a population can create different associations for different traits [19]. Thus, GWASs of crops have received widespread attention and application. Over the last decade, GWASs have been widely used in research on target trait positioning for breeding and candidate gene mining for wheat [20], rice [21,22], corn [23,24], rapeseed [25,26], soybean [27,28], sorghum [29,30], and other crops. Similarly, GWASs have also been extensively used for dissecting early maturity traits and mining candidate genes in upland cotton. Multiple GWAS methods have been utilized for association analysis of early maturity traits in upland cotton. A stable genomic region associated with the target trait was subsequently detected on chromosome D03, and the *GhAP1-D3* gene in this region was identified as having a positive regulatory effect on flowering time and early maturity [31]. Li et al. [32] performed a GWAS on early maturity via an 80 K single-nucleotide polymorphism (SNP) gene chip and mined two candidate genes, namely, *Gh-D01G0340* and *Gh-D01G1G0341*. Additionally, numerous scholars have used GWAS technology to identify SNP loci significantly associated with early maturity traits in upland cotton and have identified candidate genes related to these traits [12,33,34,35].

Although BOR is one of the most vital indicators of early maturity, few studies have focused on genetic basis determination, locus identification, and candidate gene mining for BOR in cotton. To date, more research has focused mainly on the morphology of cotton bolls [36], the boll-leaf system (BLS) [37], and the abscission of cotton boll traits [38]. However, BOR has often been overlooked for a long time due to the difficulty in identifying phenotypic traits, resulting in an unclear genetic basis, QTLs significantly associated with it, favorable allelic variations, and related genes. The lack of genetic research on BOR severely restricts the selection of early boll opening traits and the breeding of new early-maturing machine-harvested cotton varieties. Therefore, to further clarify the genetic mechanisms underlying BOR and to identify associated loci and candidate genes, a GWAS for BOR in upland cotton was conducted in this study. The results of this study provide valuable insights into the genetic basis of BOR and contribute to the breeding of new cotton varieties with improved early maturity traits suitable for mechanical harvesting. In this study, we utilized a natural population comprising 418 upland cotton accessions to identify BOR in Donghuang, Gansu (in 2020); Shihezi, Xinjiang (in 2020 and 2021); and Korla, Xinjiang (in 2020 and 2021). We subsequently utilized a mixed linear model (MLM) to perform a GWAS on SNP markers generated from whole-genome resequencing data to identify SNP loci significantly associated with BOR and relevant candidate genes, which could provide a foundation for the breeding of upland cotton through molecular-assisted selection for BOR.

## 2. Results

### 2.1. BOR Varies Widely in the Natural Population

We assessed the BOR five times in the SHZ-20 environment on 30 August, 6 September, 14 September, 20 September, and 24 September; two times in the KEL-20 environment on 24 September and 30 September; and two times in the DH-20 environment on 8 September and 21 September. Statistical analysis of the BOR values measured several times in the four environments revealed that three datasets, SHZ-20 (14 September), KEL-20 (24 September), and DH-20 (21 September), conformed to a normal distribution, and the remaining datasets tended to deviate from the normal distribution to the left or right (Appendix A). Therefore, the three normally distributed datasets were selected for subsequent analysis and provided a reference time for the collection of BOR phenotypic data in 2021.

In 2021, BOR phenotypic data were precisely acquired at SHZ-21 (14 September) and KEL-21 (24 September) (Appendix A). Therefore, we selected phenotypic data from five environments for the subsequent GWAS based on the data that best fit a normal distribution. Further analysis revealed that in the SHZ-20 environment, the distribution range of the BOR was 20.77~95.77%, the average value was 63.77%, and the coefficient of variation was 21.78%. Similarly, in the KEL-20 environment, the BOR varied substantially, ranging from 7.06% to 99.02%, with a mean of 57.17%, and the coefficient of variation was 35.64%. In the DH-20 environment, the BOR varied from 12.85% to 95.19%, with an average value of 49.10%, and the coefficient of variation was 32.04%. In the SHZ-21 environment, the BOR also varied greatly, ranging from 0.86% to 98.99%, the average value was 44.92%, and the coefficient of variation was 44.81%. In the KEL-21 environment, the distribution range of the BOR was 12.98~98.77%, with a mean of 62.57%, and the coefficient of variation was 27.81% (Table 1). Furthermore, the best linear unbiased estimation (BLUP) of the BOR ranged from 28.44% to 83.57%, with an average value of 55.48% and a coefficient of variation of 17.38%. The absolute values of the skewness values of the BOR of the individual environments and the BLUP value were between 0.13 and 0.27, and the absolute values of the kurtosis values were between 0.10 and 0.68, both of which were less than 1.00 (Table 1). Overall, the phenotypic data followed a normal distribution and varied substantially, which was extremely consistent with the basic requirements of GWASs.

Analysis of variance revealed extremely significant differences (*p* < 0.001) among the genotypes, environments, and genotype × environment interactions, which indicated that there was a significant interaction effect between the genotype and the environment on the BOR. Moreover, the generalized heritability of BOR was 80.29% (Appendix A), indicating that this trait had strong heritability and could be easily passed on to offspring. Correlation analysis revealed significant or extremely significant positive correlations among the five environments. Moreover, the correlation coefficients among the environments were between 0.1399 and 0.5787; the lowest correlation coefficient was between DH-20 and SHZ-20, with a value of 0.1399, and the greatest correlation coefficient was between KEL-21 and SHZ-21, with a value of 0.5787 (Figure 1). These results demonstrated that although the BOR of upland cotton is affected mainly by genetic factors, it can also be affected by environmental factors.

### 2.2. SNP Genotyping

To elucidate the genetic basis of BOR in-depth, a total of 418 accessions were subjected to whole-genome resequencing, and a complete set of 4,452,629 high-quality SNP markers was obtained according to the criteria of a minor allele frequency (MAF) ≥ 0.05 and a missing rate per site of less than 10%. The SNP markers consisted of 2,541,551 and 1,911,078 SNPs in the A01~A13 and D01~D13 genomes, respectively, and were unevenly distributed across all 26 chromosomes of upland cotton (Table 2; Figure 2). However, the number of SNPs significantly differed among chromosomes; chromosome A08 contained the greatest number of SNPs, with 397,496 SNPs, whereas chromosome D04 contained the lowest number of SNPs, with only 84,976 SNPs. In addition, the chromosome with the highest density of SNP distribution was A04, which contained one SNP per 1.05 kb on average, and the chromosome with the lowest density of SNP distribution was D09, with a marker density of one SNP per 3.27 kb (Table 2). Overall, the average density of the markers was approximately one SNP per 2.04 kb.

### 2.3. Population Structure and LD Analysis

To explore the number of subpopulations in the 418 upland cotton accessions, we conducted a population structure analysis based on 4,452,629 SNPs. K = 3 was considered the optimal K value in this study, indicating that the 418 accessions could be divided into three subpopulations (Figure 3A). However, most of the upland cotton materials classified into each group had mixed ancestors and did not exhibit an obvious geographical subgroup structure, suggesting that these materials might have undergone gradual introgression or extensive exchange of genes during genetic breeding, which might be the result of long-term artificial or natural selection. To further clarify the population structure for the GWAS, we conducted a principal component analysis (PCA) on the 418 accessions. The natural population materials were also divided into three groups, represented by green, blue, and orange scattered dots in Figure 3B, named G1, G2, and G3, respectively (Figure 3B). According to the clustering results, the materials of G1 and G2 contained 350 upland cotton accessions from the United States and the Yangtze River of China, whereas G3 consisted of 68 upland cotton accessions from the former Soviet Union (FSU) and the northwest region of China. Furthermore, the three groups contained multiple subgroups that were divided approximately according to geographical source, which indicated that these upland cotton accessions might have undergone gene introgression or drift during cotton breeding. A phylogenetic tree was constructed using fourfold degenerate (4D) SNPs, which are neutral or near-neutral variants, in the set of 418 accessions utilizing 4,452,629 SNPs (MAF > 0.05; Figure 3C). Notably, the phylogenetic tree contained three main subpopulations that were roughly consistent with the classification results based on phenotypic traits. Although the 418 upland cotton accessions originated from multiple pedigrees and different ecological planting regions, the natural population in this study was not highly structured, demonstrating that the natural population could be utilized for subsequent GWASs. Additionally, LD analysis was performed according to the findings of Ma et al. [33], and the average LD decay distance of the population was estimated to be approximately 187.179 kb, with *R*^2^ = 0.5 at half of the maximum value (Figure 3D).

### 2.4. GWAS for the BOR Trait in Upland Cotton

We next used the MLM algorithm to perform a GWAS based on 4,452,629 SNP markers obtained from resequencing and the BOR phenotypic values from the five individual environments and the BLUP. Here, the SNPs were considered significantly associated with −lg(*P*) values greater than 5.0, and the significance threshold was determined from previous studies [12,33]. A total of 169 significantly associated SNP loci were identified, which were located mainly on chromosomes A04, A06, A07, A09, A10, A11, D03, D05, D06, D08, D09, D11, and D12 (Appendix A; Appendix A). The MLM algorithm identified 33 SNP loci significantly associated with the BOR trait in the DH-20 environment, and these loci were distributed on chromosomes A06 and D03. Chromosome D03 contained the greatest number of SNP loci (14), and the maximum −lg(*P*) value was 6.43 for A06_116574836. In the SHZ-20 environment, only 2 SNP loci, D11_68251188 and D12_60770367, were detected on chromosomes D11 and D12, respectively. Similarly, in the KEL-21 environment, 7 SNP loci were identified on chromosomes A04, A07, D07, and D08; chromosome A04 contained the greatest number of SNP loci, with 4, and the maximum −lg(*P*) value was 5.38 for A04_11683915. However, in the KEL-20 environment, 48 loci were identified on chromosomes A07, A10, D05, D06, D09, and D11. Chromosome D11 contained the greatest number of SNP loci (41), and the maximum −lg(*P*) value was 6.07 for D11_24047649. In the SHZ-21 environment, 65 SNP loci located on chromosomes A04, A09, D09, and D11 were detected. Chromosome D11 again contained the greatest number of SNP loci (*n* = 53), and the highest −lg(*P*) value was 6.06 for D11_23741773. Finally, for BLUP, 14 SNP loci were identified on chromosomes A11, D09, and D11; chromosome D11 contained the greatest number of SNP loci with 12, and the maximum −lg(*P*) value was 5.38 for D11_23741773 (Appendix A; Appendix A). Interestingly, numerous clustered SNP loci significantly associated with BOR were identified on chromosome D11 in two environments (KEL-20 and SHZ-21) and BLUP (Appendix A; Figure 4), which indicated that there was a candidate genomic region within these QTLs on chromosome D11.

### 2.5. Identification of Candidate Genes for the BOR Trait

The GWAS results unequivocally revealed a major candidate gene region on chromosome D11 that was associated with the BOR trait. By anchoring stably associated SNP loci to chromosome D11 and detecting the corresponding potential candidate genes located in that region, we identified a significant SNP peak associated with the BOR trait between 23.703 and 23.826 Mb (Figure 5A). Further analysis revealed that a nonsynonymous mutation at the SNP locus D11_23719717 was present in this region, and the mutation SNP locus (A/G) was located in the exon of the *GH_D11G2034* gene; the gene structure is shown in Figure 5B. In addition, analysis of the protein’s three-dimensional conformational structure revealed that the mutation at this site caused significant structural changes (Figure 5C). Moreover, analysis of tissue-specific expression revealed that this gene was highly expressed in filaments, petals, and anthers (Figure 5D), suggesting that the gene may regulate cotton flowering and ultimately control BOR, determining the early maturity of cotton.

### 2.6. GH_D11G2034 May Undergo Artificial Selection in Upland Cotton

We further analyzed the SNP loci on chromosome D11 via an online website and discovered an SNP site (A/G; D11_23719717) associated with BOR in the exon of *GH_D11G2034*, and the BOR_BLUP value of materials carrying the guanine (GG) genotype was significantly greater than that of materials with the adenine (AA) genotype (Figure 6A). Thus, we inferred that GG was the favorable allelic variation (FAV) for BOR. Furthermore, the expression level of *GH_D11G2034* in accessions with early maturity with the GG genotype was significantly greater than that in late-maturing accessions with the AA genotype (Figure 6B). These results suggested that *GH_D11G2034* was differentially expressed between early- and late-maturing materials and that high levels of gene expression were beneficial for boll opening in cotton. Therefore, these findings support the theory that *GH_D11G2034* plays a crucial role in the development of early maturity.

To investigate the evolutionary models of FAV in terms of geographical origin and evolution, we compared the FAV frequency differences among the four major cotton plant regions, namely, the NIR, NSEMR, YRR, and YZRR. Initially, through comparative analysis of the BOR_BLUP values of cotton materials from the NIR, NSEMR, YRR, YZRR, FSU, and the USA, we found that the BOR of the NSEMR was significantly greater than that of the USA and other cotton regions in China (Figure 6C). However, the FAV frequencies of cotton materials from the NIR (65.4%) and NSEMR (50.0%) were greater than those from the YRR (25.2%) and YZRR (14.4%) (Figure 6D). The results demonstrated that the FAV frequencies of cotton materials in high-latitude regions were greater than those in low-latitude regions and that the frequency increased with latitude. To verify the effects of the above allelic variations on cotton boll opening traits, we selected the Min-50 and Max-50 accessions of BOR and calculated their frequency distributions. The FAV frequency of the Max-50 variety was greater than that of the Min-50 variety (Figure 6E). In addition, we estimated the π and *F*st values in neighboring regions (D11: 2,366,000–2,378,000 bp) of *GH_D11G2034.* A low π value signifies low genetic diversity and a predominance of a single nucleotide type (Figure 6F). A high *F*st value indicated that the degree of genetic differentiation and genetic differences were relatively large (Figure 6G). Thus, the results suggested that *GH-D11G2034* has been subjected to intense artificial selection throughout the process of variety selection and improvement.

## 3. Discussion

In this study, although 75.84% of the accessions were collected from different ecological cotton regions in China, we also included materials from the USA, FSU, and other countries; therefore, the phenotypes of the BOR traits in the population exhibited rich genetic variation, which provided an important basis for subsequent GWASs on the BOR traits of upland cotton. In addition, we compared cotton accessions from four different regions in China and reported that the NSEMR had a significantly greater BOR than the other three cotton regions. Although the NIR had a greater BOR than did the YZRR and YRR, the difference was not significant (Figure 6C). Therefore, we believe that NSEMR materials rather than YZRR and YRR materials should be introduced for BOR breeding in the NIR region as much as possible.

In recent years, the NIR has gradually evolved into China’s largest cotton planting region and a leading production base for high-quality cotton. Its unique geographical environment has created unique climatic conditions. For example, this region has a shorter frost-free period than those of the YRR and YZRR [39], and the annual precipitation is less than 200 mm [40]. In response to these unique climatic conditions, local agriculture frequently employs the integrated technology of subsurface drip irrigation and water fertilizer to address water scarcity and fulfill the effective accumulated temperature requirements for cotton growth and development. However, upland cotton still cannot mature naturally, and the cotton bolls do not open before machine harvesting in the NIR. In addition, the increased demand for mechanized cotton harvesting in the past decade has led to the application of defoliating and ripening agents in Chinese cotton production [41,42]. However, to spray defoliants, specific requirements for the BOR of cotton need to be met [9,10], which has resulted in a focus on artificial selection of the BOR trait in the breeding of new varieties. Therefore, after years of artificial selection and domestication, the BOR of upland cotton varieties in the NIR region was significantly greater than that in other regions.

Although the GWAS technique has been widely applied in the fields of cotton yield [43,44], quality [43,45], plant architecture [46,47], and resistance to *Verticillium* wilt [48] and some GWASs have included early maturity-related traits [12,31,32], few studies on BOR, one of the most important indicators for measuring early maturity, have been performed. In this study, we used an MLM [32,49] with a GWAS based on resequencing data to explore favorable alleles of BOR to improve target traits in cotton production. Here, significant trait-associated SNPs were defined as those with −lg(*P*) values higher than 5.0, and the significance threshold was determined from previous studies [12,33]. Thus, a total of 169 significantly associated SNP loci were largely located on chromosomes A04, A06, A07, A09, A10, A11, D03, D05, D06, D08, D09, D11, and D12 (Appendix A and Appendix A). Furthermore, we used only significant SNPs that were detected in at least two environments to obtain more reliable results [32]. Ultimately, we identified 18 significant SNPs associated with BOR traits. Notably, these SNPs were consistently located on chromosome D11, suggesting the possibility that this region is a genomic candidate region associated with the target trait on chromosome D11. Although previous studies have shown that chromosome D03 is rich in QTLs for early maturity traits [12,31,41,50,51], we identified a genomic region on chromosome D11 that was significantly and stably associated with BOR, which might constitute a potential QTL interval related to the early maturity traits of upland cotton.

The identification of stable QTLs and potential candidate genes can provide valuable reference information for marker-assisted selection (MAS) breeding [32]. Previous studies have shown that mutation and accumulation of favorable alleles may be the two main ways to improve important target traits in crops [12,52]. In this study, the GWAS results strongly suggested that chromosome D11 was a major region with potential candidate genes related to the BOR trait. Furthermore, we identified a significant SNP peak associated with the BOR trait between 23.703 and 23.826 Mb, in which a nonsynonymous mutation SNP locus (A/G) of D11_23719717 was identified; the mutation SNP locus was located in the exon of the *GH_D11G2034* gene. In addition, we found that the BOR_BLUP of accessions with the GG genotype was significantly greater than that of accessions with the AA genotype (Figure 6A), which indicated that the GG genotype was the FAV for BOR. Moreover, the FAV frequency of cotton varieties in high-latitude regions was greater than that in low-latitude regions; the NIR presented the highest FAV frequency, which demonstrated that *GH_D11G2034* may play a crucial role in the BOR-focused breeding of upland cotton. The analysis of π and *F*st indicated that *GH_D11G2034* was a highly elite allele for the early maturity of cotton. Additionally, *GH-D11G2034* has undergone strong artificial selection in the process of cotton variety breeding. The above results provide important genetic resources and a reference basis for the molecular breeding of BOR and other early-maturing traits in upland cotton.

## 4. Materials and Methods

### 4.1. Experimental Materials

In this study, a total of 418 upland cotton accessions were collected, all of which presented genetic characteristics through several generations of self-pollination (Appendix A). The germplasm included in this population is the core germplasm resource of Chinese upland cotton, with extensive genetic variation and representativeness. This study was approved by the Institutional Review Board (IRB) of the Xinjiang Academy of Agricultural Sciences. The accessions were divided into six groups: the FSU group (*n* = 25), the USA group (*n* = 57), the NIR group (*n* = 28), the NSEMR group (*n* = 14), the YRR group (*n* = 168), the YZRR group (*n* = 107) and others (*n* = 19). We subsequently used the 418 accessions to construct an association mapping panel for GWAS analysis.

### 4.2. Field Experimental Design

In 2020, a total of 418 accessions were planted in the experimental field of the Cotton Research Institute of the Xinjiang Academy of Agricultural Reclamation Sciences in Shihezi, Xinjiang; the Kuerle Experimental Station of the Xinjiang Academy of Agricultural Reclamation Sciences in Kuerle, Xinjiang; and the Dunhuang Cotton Experimental Station of the Gansu Academy of Agricultural Sciences in Dunhuang, Gansu, named SHZ-20, KEL-20 and DH-20, respectively. In 2021, a total of 418 accessions were planted in the experimental fields of the Cotton Research Institute of the Xinjiang Academy of Agricultural Reclamation Sciences in Shihezi, Xinjiang, and at the Kuerle Experimental Station of the Xinjiang Academy of Agricultural Reclamation Sciences in Kuerle, Xinjiang; these accessions were named SHZ-21 and KEL-21, respectively.

Field experiments were conducted in six experimental environments via a randomized block arrangement, with three replicates for each planting environment. For DH-20, the plant spacing was set at 0.15 m, the row spacing at 0.40 m, the width at 1.45 m, the film surface at 1.20 m, and the film spacing at 0.40 m. Each replicate contained approximately 35 individual plants of each material. In the other environments, each material was planted in two rows with a row spacing of 0.66 m and a spacing of 0.10 m. Each material had approximately 40 plants per repetition, and field management in each environment was carried out according to local conventional methods.

### 4.3. Statistical Analysis of Phenotypic Data

After the normal boll opening of the materials at each test site, we selected ten individual plants with neat growth and approximately uniform growth for each repetition of each material and determined the total number of cotton bolls and the number of opening bolls of the ten individual plants (only bolls with a diameter of more than 2.0 cm on a single plant were counted, and the remaining bolls were ignored). The BOR of each accession was calculated via the following formula: BOR = (boll opening number/total boll number) × 100%. Excel 2021 and IBM SPSS Statistics 26.0 software were used to conduct descriptive statistics and variance analyses on the phenotypic values of BOR in each environment. The basic parameters of the statistical analysis were the coefficient of variation (CV), standard deviation (SD), minimum (Min), maximum (Max), and mean. Additionally, software such as Origin 2018, Adobe Photoshop CS6, and Adobe Illustrator 2018 were used to draw and combine the images. The BLUP was calculated via the R package “lme4” [53]. The formula for calculating broad-sense heritability (*H*^2^) was *H*^2^ = ${\sigma_{g}}^{2}({{{\sigma_{g}}^{2}+\sigma }_{gy}}^{2}/n+{\sigma_{e}}^{2}/nr)$. where ${\sigma_{g}}^{2},{\sigma_{gy}}^{2}$ and ${\sigma_{e}}^{2}$ represent genetic variance, genotype and environmental interaction (G × E) variance and error variance, respectively, and ‘n’ and ‘r’ represent the number of environments and replicates, respectively [54].

### 4.4. SNP Identification

All the raw data and sequence information of 418 accessions were downloaded from the article by Ma et al. [33]. For each accession, clean reads were first aligned to the most recent genome sequence (*G. hirsutum* acc. TM-1) [55] via the BWA-MEM algorithm with the Burrows-Wheeler Aligner (version v0.7.17), and SNP calling was performed via SAMtools-V1.17 [56]. Next, we used the software VCFtools V0.1.13 [57] for genotypic filtering to remove SNP markers with MAFs greater than 5% and missing rates less than 10%.

### 4.5. Population Structure and LD Analyses

In this study, we identified 4,452,629 high-quality SNPs with an MAF of 0.05 and a missing rate per site of less than 10% within the population for population structure and LD analyses. The population structure was investigated with ADMIXTURE V1.3.0 [58,59], and a phylogenetic tree was drawn and visualized via SNPhylo [60] and visualized via the R package ‘ggtree’ [61]. The software GCTA (version 1.26.0) [48,62] was used for PCA, and LD was calculated via PLINK (version 1.9) [63,64].

### 4.6. GWAS for BOR in Upland Cotton

A total of 4,452,629 high-quality SNPs were utilized to perform a GWAS for BOR in 418 accessions via the MLM algorithm in the TASSEL version 3.0 software package [49]. Using the ‘lme4’ R package [53], we obtained the BLUP BOR values among 418 accessions. The significance threshold P was determined with reference to Ma et al. [33] and Li et al. [12], and SNP loci with −lg(*P*) ≥ 5.00 were considered significantly associated target traits. Finally, the Manhattan plot and Q‒Q heatmap were constructed with the qqman package of R 4.4.2 [65].

### 4.7. Identification of Elite Alleles and Candidate Genes for the BOR Trait

SNP loci that could be detected simultaneously in multiple environments were considered significantly associated with BOR. IBM SPSS Statistics 26.0 was used to collate the average values of the phenotypes of the variations of significantly associated loci and to conduct two-tailed *t* tests to determine the specific elite allele variations of a single SNP locus. The corresponding box chart was drawn via Origin 2018. According to the GWAS results, we compared the physical location of the significant SNP loci with the physical location of the sequences in the cotton genome database; these sequences were extended 200 kb upstream and downstream and defined as candidate genomic regions. We subsequently used gene function annotation information from cotton genome websites (http://cotton.zju.edu.cn/index.htm, accessed on 18 October 2024) to identify candidate genes for the BOR trait.

## 5. Conclusions

In this study, we used the MLM algorithm to perform a GWAS based on 4,452,629 SNP markers obtained from resequencing and BOR phenotypic values from five individual environments and the BLUP. A total of 169 significantly associated SNP loci were largely located on chromosomes A04, A06, A07, A09, A10, A11, D03, D05, D06, D08, D09, D11, and D12. Furthermore, we used only significant SNPs that were detected in at least two environments to obtain more reliable results. Ultimately, a total of 18 stable significant SNP loci associated with BOR were located on chromosome D11 and were identified in more than one environment. Furthermore, we identified a significant SNP peak (23.703–23.826 Mb) associated with the BOR trait. A nonsynonymous mutation SNP locus (A/G) at D11_23719717 was identified in this region, and the mutation SNP locus was located in the exon of the *GH_D11G2034* gene. Furthermore, our study revealed that an FAV (GG) in *GH-D11G2034* was significantly correlated with BOR, and high expression levels of *GH-D11G2034* were beneficial for cotton boll opening. Evolutionary analysis revealed that *GH-D11G2034* underwent strong artificial selection throughout the variety selection process. In addition, tissue-specific expression analysis revealed that the gene was highly expressed in filaments, petals, and anthers, indicating that *GH-D11G2034* may regulate cotton flowering and control the boll opening rate, ultimately determining cotton BOR and early maturity.

## Figures and Tables

**Figure 1 ijms-26-02697-f001:**
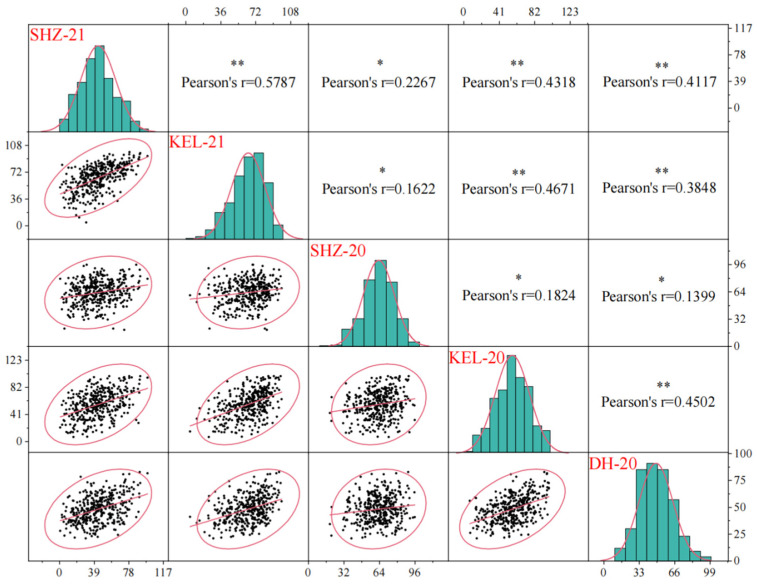
Trait correlations between environments. The diagonal position of each trait is the frequency distribution histogram of the environment. The lower left corner of the diagonal is the scatterplot of floc production in the two environments. The upper right corner of the diagonal line is the correlation coefficient for BOR among the environments. ** indicates a significant correlation at *p* < 0.01. * represents a significant correlation at *p* < 0.05.

**Figure 2 ijms-26-02697-f002:**
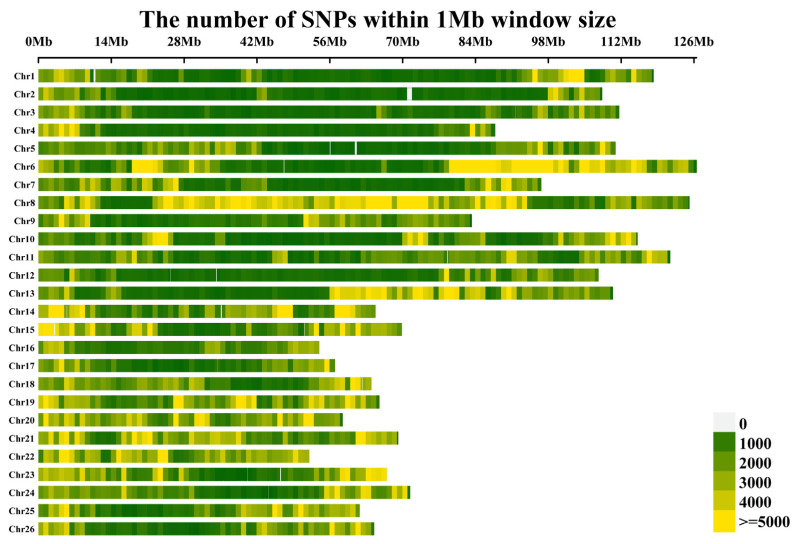
Single-nucleotide polymorphism (SNP) distributions across the 26 chromosomes (A01–A13; D01–D13) of the upland cotton genome.

**Figure 3 ijms-26-02697-f003:**
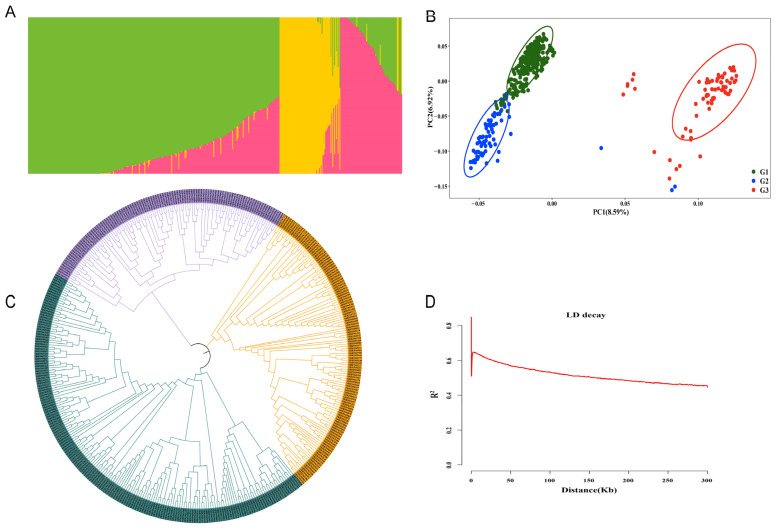
Population structure analysis. (**A**) The total cohort was grouped into 3 subpopulations (K = 3); each color represents one subpopulation. (**B**) PCA plots of 418 accessions. (**C**) Phylogenetic tree of 418 accessions inferred from 4,452,629 SNPs at fourfold degenerate sites, including three groups. (**D**) The LD decay curve was determined according to *r*^2^.

**Figure 4 ijms-26-02697-f004:**
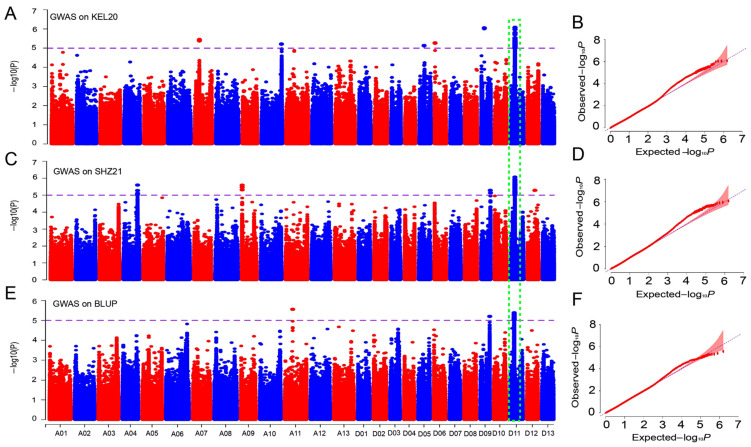
Numerous clustered SNP loci significantly associated with BOR were identified on chromosome D11 in two environments (KEL-20 and SHZ-21) and in BLUP. (**A**,**B**), Manhattan and Q‒Q plots for KEL-20. (**C**,**D**), Manhattan and Q‒Q plots for SHZ-21. (**E**,**F**), Manhattan and Q‒Q plots for BLUP.

**Figure 5 ijms-26-02697-f005:**
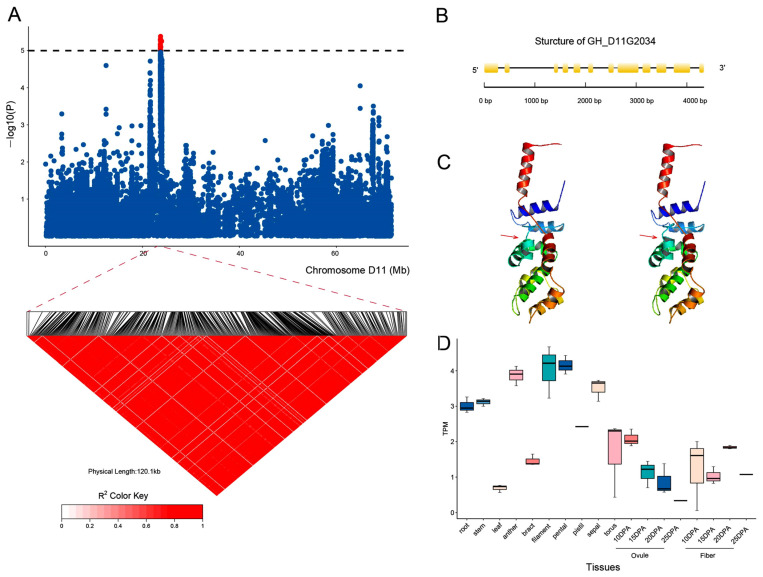
GWAS for BOR and identification of the candidate gene *GH_D11G2034* on chromosome D11. (**A**) Manhattan plots for BOR on chromosome D11; arrowheads indicate the strong D11_23719717 locus on the candidate gene *GH_D11G2034*. (**B**) Genetic structure of *GH_D11G2034*. Yellow and black represent exons and introns, respectively. (**C**) Comparison of the expression levels of *GH_D11G2034* in different cotton tissues. (**D**) Changes in three-dimensional protein conformation caused by alterations in the SNP locus. The red arrowheads indicate the specific location of the change.

**Figure 6 ijms-26-02697-f006:**
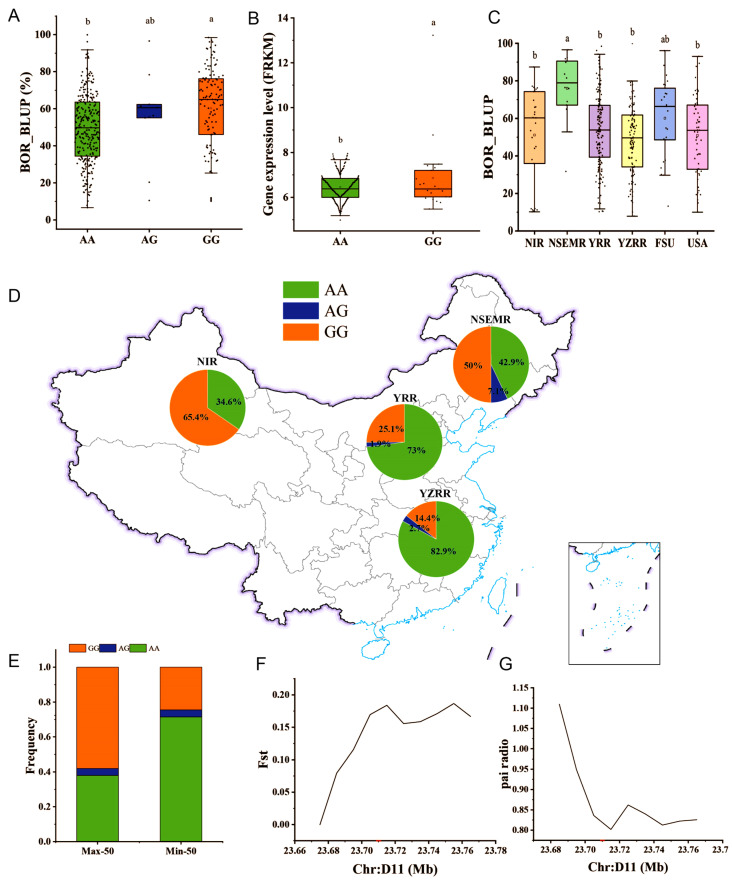
Identification of excellent allelic variation in *GH_D11G2034*. (**A**) Differences in BOR_ BLUP among 418 upland cotton accessions. (**B**) *GH_D11G2034* expression data from early- and late-maturing accessions. (**C**) Comparison of differences in BOR_BLUP among four different Chinese cotton regions [NIR, northern super early maturity region (NSEMR), YRR, YZRR], the former Soviet Union (FSU), and the United States of America (USA). Different lowercase letters in (**A**–**C**) indicate significant differences (*p* < 0.05). (**D**) Geographical distribution frequency of FAVs in the four Chinese cotton regions. (**E**) Frequency of FAV in Min-50 and Max-50 high- and low-BOR_BLUP extreme accessions. (**F**) Genetic differentiation index (*F*st) calculation. (**G**) Nucleotide polymorphism (π) ratio in the population with a Min-50 to Max-50 of BOR_BLUP extreme accessions.

**Table 1 ijms-26-02697-t001:** Descriptive statistics of BOR phenotypic data.

Environments ^a^	Max (%)	Min (%)	Mean (%)	SD ^b^	CV ^c^ (%)	Skewness	Kurtosis
SHZ-20	95.77	20.77	63.77	13.89	21.78	−0.27	−0.23
KEL-20	99.02	7.06	57.12	20.36	35.64	−0.13	−0.52
DH-20	95.19	12.85	49.1	15.73	32.04	0.26	−0.18
SHZ-21	98.99	0.86	44.92	20.13	44.81	0.23	−0.42
KEL-21	98.77	12.98	62.57	17.4	27.81	−0.17	−0.68
BLUP	83.57	28.44	55.48	9.64	17.38	0.2	−0.1

^a^ SHZ-20, KEL-20, DH-20, SHZ-21, KEL-21 and BLUP stand for 2020 in Shihezi, Xinjiang; 2020 in Kuerle, Xinjiang; 2020 in Donghuang; 2021 in Shihezi; 2021 in Kuerle; and best linear unbiased prediction (BLUP). ^b^ SD, standard deviation; ^c^ CV, coefficient of variation.

**Table 2 ijms-26-02697-t002:** SNP distribution across chromosomes.

Chr.	SNP Number	Chr Length (Mb)	SNP Density (Kb)	Chr.	SNP Number	Chr. Length (Mb)	SNP Density (Kb)
A01	174,575	118.15	1.48	D01	183,419	64.70	2.84
A02	122,573	108.27	1.13	D02	179,130	69.78	2.57
A03	245,146	111.59	2.2	D03	94,075	53.9	1.75
A04	91,660	87.7	1.05	D04	84,976	56.93	1.49
A05	162,903	110.84	1.47	D05	141,897	63.92	2.22
A06	351,101	126.49	2.78	D06	177,358	65.46	2.71
A07	144,608	96.6	1.5	D07	169,896	58.42	2.91
A08	397,496	125.06	3.18	D08	183,719	69.08	2.66
A09	140,054	83.22	1.68	D09	169,952	52.00	3.27
A10	177,383	115.1	1.54	D10	143,111	66.88	2.14
A11	184,666	121.38	1.52	D11	130,046	71.36	1.82
A12	119,840	107.57	1.11	D12	136,328	61.69	2.21
A13	229,546	110.37	2.08	D13	117,171	64.45	1.82
A01~A13	2,541,551			D01~D13	1,911,078		

## Data Availability

All the data generated or analyzed during this study are included in this article and its Appendix A.

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
