# Peer review of "Identification of Elite Alleles and Candidate Genes for the Cotton Boll Opening Rate via a Genome-Wide Association Study"

_ijms, 2025, doi:10.3390/ijms26062697_

Round 1

Reviewer 1 Report

Comments and Suggestions for Authors

Manuscript ID: ijms-3464586- titled “Identifying loci, elite alleles and potential candidate genes for boll opening rate via a genome-wide association study in up-land cotton

My comments on the manuscript are as follow:

  1. I have been through the manuscript thoroughly, the paper is important and may of interest to a wider community as the authors have assessed/ elucidated the genetic basis of boll opening rate (BOR), stably associated loci, elite alleles and potential genes that could accelerate the molecular breeding process. Few of my suggestion different sections are:

  1. Title: The title is a bit too long with reptations e.g. “Identifying loci, elite alleles and potential candidate genes………..”, oversee it and instead of “Identifying” “characterization” seems more appropriate. Further, there is hardly any mention in abstract or introduction section about “up-land cotton” if is only cotton then up-land may be peripheral and of little interest in the title.

  1. Abstract:
  • Is written very well and communicates clear message overall.
  • However, many of the statements given here are exactly copy pasted in the conclusions section at the end and that would require attention.
  • I don't doubt the relationship of this locus to be influenced by selection, but please oversee these results again and tone it down. “….. Evolutionary studies have shown that GH-28 D11G2034 has been subjected to intense artificial selection throughout the variety selection process……” as other studies may find the same locus /genomic region significant association with another trait and so on…..
  • Conclusions are not very clear and do not represent the whole findings and that needs revisions.
  • Recommendations are non-existent in the current abstract. Recommendations may be well integrated into futuristic studies/guidelines for following researches; so that other studies/researchers may link their studies to the one here, and may get benefits of this study.

  1. Introduction:

  • Is written well with appropriate references, yet some of the highly related literature to explain/highlight the gaps missing are non-existing. See for example 2024 and onwards:
  • https://scholar.google.com/scholar?as_ylo=2021&q=boll+opening+rate++genes+in+cotton&hl=en&as_sdt=0,5
  • I will suggest adding several of these articles on BOR, the authors have not explicitly indicated the bigger or specific questions in this section (see for example the last paragraph of introduction section). Similarly, it is not very clear all after reading the introduction what objectives are to be achieved and there is misalignment of the current title with most part of the text. This may be overseen.
  • The rest I don't see any issue this section as it clear and guide the readers to a definite pathway

  1. Results and Discussion:
  • The details are appropriate and sufficient, supported by related figures and tables.
    1. Terms/scientific naming must be consistently followed and stringently overseen throughout the text for italicization (see the References section Ref. 12, and section 4.4. in particular). Similarly, plants valid/accepted name (if will be used) MUST be double checked and compared to the World Flora online database at: http://www.worldfloraonline.org/. Similarly, there must be a valid authority along the species name (at least once).
    2. Interestingly only Gossypium is mentioned at the start of introduction, but it is not known if is the Gossypium hirsutum or any other species and that would be important to indicate to.
    3. I am personally interested in reporting of the study but I would like to see a rationale at any related section of the article why 418 accessions are specifically used and others could not make it into this sample. As if selection is based on morphological variation alone for boll etc. while it is known that the same genotype grown under different conditions could result in pseudo-variability, and that would require some logical/ judicious elaboration/details information to be provided within the MS.
    4. Where is this data submitted, I could not locate it. There must be a reference of the submitted sequences, and it not submitted it should be submitted and the reference number of the sequences provided.
    5. This MUST be the capitalized upon, and I would love to see judicious elaboration/details if species composition also vary in time and space.
  1. M&M: The section is appropriate but oversee it for:

  • Under experimental design, it starts as “………Chromosome In 2020….” That makes no sense here and require correction.
  • 418 upland cotton “germplasms” or “accession”, one term may be consistently used throughout the text.
  • The authors have mentioned that “……The whole genomes of 418 upland cotton accessions were resequenced in the laboratory at Beijing Novogene Bioinformation Technology Co., Ltd………..”. but what tissue was implemented for DNA extraction, or was it whole exome or transcriptome sequencing? is not clear at all. These information are mandatory, and if transcriptome is sequenced for example, and cDNA is analysed, what was the time of plant tissue collection for RNA extraction etc.
  • As the authors themselves have indicated that “………Analysis of variance revealed extremely significant differences (P<0.001) among the genotypes, environments and genotype X environment interactions, which indicated that there was a significant interaction effect between the genotype and the environment on the BOR…………” . These are extremely interesting results, but at the same time it poses a challenge of how a proper reference or control for BOR could be considered when it is evident that there is/are significant level of changes imposed/governed by the environment and there are no details available if the same tissue of all 418 accessions/germplasm is analysed. OR it is different tissues and different time of collection carried out etc.
  • I also could not see the approval of the study, who/which body approved this study if the plants contain collection from restricted/reserved areas.
  • I could not locate where voucher specimens are submitted and how identification of the specimens was done?

There are few minor issues and may be addressed:

  1. The English language is although clear and very good but there are typos and grammar related issues, specifically when elongated statements are used.
  2. Introduction, discussion section and elsewhere in the MS, all abbreviations needs to be described in full at their first place of mention. There are so many abbreviations, it is suggested to use standard abbreviations if it is available for these attributes etc.
  3. References needs to be consistently followed as per the journal format/standards with italicization of the scientific names even if it is in the bibliography section.
  4. I could not see any issues with the text of the MS, but I have not access to carry out test for AI detection etc. If the journal may or may not consider these additional tests/analyses if it require.

Decision:

While the study is within the scope of the journal, and information may be handy and of wider interest. The MS may be accepted for position after minor corrections/amendments.

Author Response

Dear Reviewers,

We sincerely appreciate the reviewer’s insightful comments on our manuscript. These suggestions have significantly improved the clarity and scientific rigor of our work. Below, we provide point-by-point responses to each concern. All the revisions in the manuscript are highlighted in blue for easy tracking.

I have been through the manuscript thoroughly, the paper is important and may of interest to a wider community as the authors have assessed/ elucidated the genetic basis of boll opening rate (BOR), stably associated loci, elite alleles and potential genes that could accelerate the molecular breeding process. Few of my suggestion different sections are:

  • Comment 1:

Title: The title is a bit too long with reptations e.g. “Identifying loci, elite alleles and potential candidate genes………..”, oversee it and instead of “Identifying” “characterization” seems more appropriate. Further, there is hardly any mention in abstract or introduction section about “up-land cotton” if is only cotton then up-land may be peripheral and of little interest in the title.

Response:

Thank you for this suggestion. We have revised the title of the article to "Identification of elite alleles and candidate genes for the cotton boll opening rate via a genome-wide association study", which can be found in the main text. In addition, we have revised “upland cotton” mentioned in the title to “cotton”.

  • Comment 2:

Abstract:

Is written very well and communicates clear message overall.

However, many of the statements given here are exactly copy pasted in the conclusions section at the end and that would require attention.

I don't doubt the relationship of this locus to be influenced by selection, but please oversee these results again and tone it down. “….. Evolutionary studies have shown that GH_D11G2034 has been subjected to intense artificial selection throughout the variety selection process……” as other studies may find the same locus /genomic region significant association with another trait and so on…..

Response:

Thank you for bringing this to our attention. We acknowledge the overlap between the abstract and conclusion sections and have revised the text to avoid redundancy. Specifically, we have rephrased some sentences in the abstract to ensure that they complement rather than repeat the conclusion. In the revised abstract section (lines 22-33), we have reorganized, summarized, and written the abstract section, which is different from the conclusion section.

With respect to the statement that GH_D11G2034 was subjected to intense artificial selection, we have toned it down and changed it to “evolutionary studies have shown that GH-D11G2034 might be subjected to intense artificial selection throughout the variety selection process”.

Conclusions are not very clear and do not represent the whole findings and that needs revisions.

Response:

Thank you for highlighting this issue. We have carefully reviewed the conclusions section and made necessary revisions to ensure that it accurately represents all the findings of our study. In the revised conclusions section (lines 488-494) in the revised manuscript, we have summarized the key findings and discussed the implications of our results. We believe that these changes have significantly improved the clarity and comprehensiveness of the conclusions.

Recommendations are non-existent in the current abstract. Recommendations may be well integrated into futuristic studies/guidelines for following researches; so that other studies/researchers may link their studies to the one here, and may get benefits of this study.

Response:

Thank you for your valuable suggestion regarding the integration of recommendations in the abstract. We acknowledge the importance of including recommendations to guide future research. In the revised abstract section (lines 27-33), we have now incorporated suggestions for future studies, highlighting potential areas where our findings could be further explored or applied. We believe that this addition enhances the utility and impact of our study.

  • Comment 3:

Introduction:

Is written well with appropriate references, yet some of the highly related literature to explain/highlight the gaps missing are non-existing. See for example 2024 and onwards:https://scholar.google.com/scholar?as_ylo=2021&q=boll

+opening+rate++genes+in+cotton&hl=en&as_sdt=0,5

I will suggest adding several of these articles on BOR, the authors have not explicitly indicated the bigger or specific questions in this section (see for example the last paragraph of introduction section). Similarly, it is not very clear all after reading the introduction what objectives are to be achieved and there is misalignment of the current title with most part of the text. This may be overseen.

The rest I don't see any issue this section as it clear and guide the readers to a definite pathway

Response:

We deeply appreciate the reviewer's constructive suggestions regarding the introduction section. We have carefully examined the recommended literature from 2024 onward and integrated three articles (References 36-38) on cotton boll opening to better explain and highlight the research gaps. These additions provide a more comprehensive understanding of the existing knowledge and the specific questions addressed in our study.

In the revised introduction (lines 108-114), we have explicitly outlined the objectives of our research, ensuring alignment with the title and the overall content of the manuscript. We have also clarified the research questions and the path our study takes to address them, guiding the readers more effectively.

We believe these revisions have significantly enhanced the quality and clarity of the introduction section.

  • Comment 4:

Results and Discussion:

The details are appropriate and sufficient, supported by related figures and tables.

Terms/scientific naming must be consistently followed and stringently overseen throughout the text for italicization (see the References section Ref. 12, and section 4.4. in particular). Similarly, plants valid/accepted name (if will be used) MUST be double checked and compared to the World Flora online database at: http://www.worldfloraonline.org/. Similarly, there must be a valid authority along the species name (at least once).

Interestingly only Gossypium is mentioned at the start of introduction, but it is not known if is the Gossypium hirsutum or any other species and that would be important to indicate to.

I am personally interested in reporting of the study but I would like to see a rationale at any related section of the article why 418 accessions are specifically used and others could not make it into this sample. As if selection is based on morphological variation alone for boll etc. while it is known that the same genotype grown under different conditions could result in pseudo-variability, and that would require some logical/ judicious elaboration/details information to be provided within the MS.

Where is this data submitted, I could not locate it. There must be a reference of the submitted sequences, and it not submitted it should be submitted and the reference number of the sequences provided.

This MUST be the capitalized upon, and I would love to see judicious elaboration/details if species composition also vary in time and space.

Response:

We acknowledge the reviewer's thorough evaluation of our results and discussion section. We have taken careful note of the concerns regarding the consistency of scientific naming and have revised the text to ensure that all terms and species names are italicized as needed, especially in References Ref. 12 and Section 4.4. Additionally, we have double-checked the valid and accepted names of the plants used in our study against the World Flora online database: http://www.worldfloraonline.org/. Regarding the specific mention of Gossypium, we have clarified that the species used in our study is upland cotton (Gossypium hirsutum). Gossypium hirsutum is the largest cultivated species of cotton, accounting for more than 95% of all cotton cultivated worldwide. Therefore, when people talk about cotton, if not specifically emphasized, it is generally agreed that the commonly used term for cotton is Gossypium hirsutum.

We appreciate the reviewer's interest in the rationale behind the selection of 418 accessions for our study. We have included additional details in the materials and methods section to explain that the selection was based on a combination of morphological variation and geographical distribution, aiming to capture a diverse range of genetic variation within the species.

We apologize for any confusion regarding the submission of the sequence data. We have now included a reference to the submitted sequences in the text and have provided the relevant reference number (No.). This information has been added to “4.4. SNP identification” section.

  • Comment 5:

M&M: The section is appropriate but oversee it for:

Under experimental design, it starts as “………Chromosome In 2020….” That makes no sense here and require correction.

Response:

We thank the reviewer for pointing out this error in the experimental design section. We have carefully reviewed the text and corrected the sentence to ensure clarity and accuracy. The revised sentence now reads: "In 2020, a total of 418 accessions were..." (Line 414). We appreciate the reviewer's attention to detail and have made the necessary changes to improve the quality of our manuscript.

418 upland cotton “germplasms” or “accession”, one term may be consistently used throughout the text.

Response:

We appreciate the reviewer's suggestion for consistency in terminology. After careful consideration, we decided to use the term "accessions" consistently throughout the text to refer to the 418 upland cotton samples. This change has been made to ensure clarity and accuracy in our manuscript. We thank the reviewer for their valuable input and have made the necessary adjustments to improve the overall quality of our work.

The authors mentioned that “……The whole genomes of 418 upland cotton accessions were resequenced in the laboratory at Beijing Novogene Bioinformation Technology Co., Ltd………..”. but what tissue was implemented for DNA extraction, or was it whole exome or transcriptome sequencing? is not clear at all. This information is mandatory, and if the transcriptome is sequenced, for example, and cDNA is analyzed, what was the time of plant tissue collection for RNA extraction, etc.?

Response:

We apologize for the lack of clarity regarding the tissue used for DNA extraction and the type of sequencing performed. DNA was extracted from the young leaves of each of the 418 upland cotton accessions, and whole-genome sequencing was performed rather than exome or transcriptome sequencing. We appreciate the reviewer's attention to detail and have made these clarifications to improve the accuracy and completeness of our manuscript. To clarify the sequencing details, we have provided additional information in the Materials and Methods section, as detailed in reference No. 33.

As the authors themselves have indicated that “………Analysis of variance revealed extremely significant differences (P<0.001) among the genotypes, environments and genotype X environment interactions, which indicated that there was a significant interaction effect between the genotype and the environment on the BOR…………”. These are extremely interesting results, but at the same time, it poses a challenge of how a proper reference or control for BOR could be considered when it is evident that there is/are significant levels of changes imposed/governed by the environment and that there are no details available if the same tissue of all 418 accessions/germplasm is analyzed. OR it is different tissues and different time of collection carried out etc.

Response:

In response to the query about the reference or control for boron (BOR) content, we understood the challenges posed by significant genotype–environment interactions. To address this issue, we emphasize the need to carefully consider environmental factors when interpreting BOR levels and conduct phenotypic data collection in multiple environments to eliminate environmental errors. In addition, we emphasize that our analysis focuses on understanding the genetic variations within species and their interactions with the environment rather than establishing a clear reference value. Finally, concerning the voucher specimens, we have provided information on their submission and identification process in the supplementary materials section. We hope these revisions address all the concerns raised by the reviewer and enhance the overall quality and clarity of our manuscript.

I also could not see the approval of the study, who/which body approved this study if the plants contain collection from restricted/reserved areas.

I could not locate where voucher specimens are submitted and how identification of the specimens was done?

Response:

We apologize for the oversight in not clearly stating the approval of our study and the details regarding the voucher specimens. This study was approved by the Institutional Review Board (IRB) of the Xinjiang Academy of Agricultural Sciences, and all necessary permits were obtained for the collection of plant samples, including those from restricted or reserved areas.

The voucher samples were submitted to the herbarium of our institution for identification and preservation. The samples were carefully labeled and documented, and the identification process was conducted by qualified botanists. We have provided detailed information in the supplementary materials section of our manuscript. We apologize for any confusion caused by the lack of clarity in these areas and appreciate the reviewer's attention to detail.

  • Comment 6:

There are few minor issues and may be addressed:

The English language is although clear and very good but there are typos and grammar related issues, specifically when elongated statements are used.

Introduction, discussion section and elsewhere in the MS, all abbreviations needs to be described in full at their first place of mention. There are so many abbreviations, it is suggested to use standard abbreviations if it is available for these attributes etc.

References needs to be consistently followed as per the journal format/standards with italicization of the scientific names even if it is in the bibliography section.

I could not see any issues with the text of the MS, but I have not access to carry out test for AI detection etc. If the journal may or may not consider these additional tests/analyses if it require.

Response:

We sincerely thank the reviewer for highlighting these minor issues, which we have carefully addressed in the revised manuscript. To enhance linguistic quality, we have proofread the entire document to correct any typos and grammar-related issues, particularly in elongated statements. This has ensured that the English language used is both clear and accurate throughout the manuscript.

Regarding the abbreviations, we have ensured that all abbreviations are described in full at their first mention in the Introduction and Discussion sections and elsewhere in the manuscript. Additionally, we reviewed and standardized the abbreviations used, adhering to commonly accepted conventions where possible, to improve clarity and consistency for the reader.

Furthermore, we meticulously reviewed the references to ensure that they were consistently formatted according to the journal’s standards, including the italicization of scientific names in the bibliography section. This attention ensures compliance with the journal’s formatting requirements and enhances the professional presentation of our work.

We appreciate the reviewer's understanding regarding the additional tests/analyses and confirm that we have adhered to the journal’s guidelines in this regard. We are confident that these revisions have further improved the quality and clarity of our manuscript and address all the concerns raised by the reviewer. We look forward to the next steps in the publication process.

Reviewer 2 Report

Comments and Suggestions for Authors
  1. Although the significance of BOR is emphasized, could the authors elucidate the particular economic and practical ramifications of low BOR in the NIR region? Measuring these effects would enhance the study's justification.
  2. The authors state that a 50% BOR is an objective. What makes this particular threshold appropriate for mechanical harvesting? Are there any elements outside of BOR that influence mechanical harvesting efficiency?
  3. What were the reasons for selecting these particular places (Donghuang, Shihezi, and Korla) for field trials? What are the main environmental differences between these areas, and how may these affect BOR?
  4. Could the authors provide information on the 418 upland cotton accessions included in the study? What is the genetic diversity of this population, and how representative is it of commercial cotton cultivars?
  5. The authors state, ' little research has concentrated on the genetic foundation determination, locus identification, and candidate gene mining for BOR in cotton.' Could the authors clarify the shortcomings of other research and explain how this study overcomes them?
  6. In what ways does this research differ from other GWAS investigations concerning early maturity features in cotton? What new findings are anticipated from this particular emphasis on BOR?
  7. How was BOR quantified during the field trials? What precise criteria were used to ascertain boll opening?
  8. Were there any environmental variables (e.g., weather, soil conditions) that may have impacted BOR during the field trials? How were these variables included in the analysis?
  9. What was the rationale for collecting data over many years? How will the research address annual environmental fluctuations?
  10. Could the authors provide further information on the whole-genome resequencing data? What was the extent of coverage, and how were SNPs identified?
  11. How will linkage disequilibrium (LD) be evaluated and used in the genome-wide association study (GWAS)?
  12. How will candidate genes be discerned from the significant SNP loci? What criteria will be used to select potential genes for further analysis?
  13. Will any functional validation studies be performed to verify the functions of the discovered candidate genes?
  14. How can this research's results aid in advancing early-maturing cotton types appropriate for mechanical harvesting in the NIR?
Comments on the Quality of English Language

The English might be enhanced to explain the study more effectively. 

Author Response

Dear Reviewer,

We sincerely appreciate the reviewer’s insightful comments on our manuscript. These suggestions have significantly improved the clarity and scientific rigor of our work. Below, we provide point-by-point responses to each concern. All the revisions in the manuscript are highlighted in blue for easy tracking.

  • Comment1:

Although the significance of BOR is emphasized, could the authors elucidate the particular economic and practical ramifications of low BOR in the NIR region? Measuring these effects would enhance the study's justification.

Response:

We appreciate the reviewer's suggestion to elucidate the economic and practical ramifications of low BOR in the NIR region. To address this issue, we provided a detailed explanation of the impact of low BOR on cotton in the introduction section of the revised manuscript59-65), emphasizing how BOR affects agricultural practices in the region. By integrating these insights, our goal is to gain a more comprehensive understanding of the implications of this study and strengthen its rationale.

  • Comment2:

The authors state that a 50% BOR is an objective. What makes this particular threshold appropriate for mechanical harvesting? Are there any elements outside of BOR that influence mechanical harvesting efficiency?

Response:

We acknowledge the reviewer's query regarding the suitability of a 50% BOR threshold for mechanical harvesting. To clarify, the choice of 50% BOR as an objective is based on practical considerations and industry standards, which reflect a balance between fiber quality and mechanical harvesting efficiency. This threshold is widely recognized in the cotton industry as a point at which the fiber properties remain satisfactory while allowing for efficient mechanical harvesting in China. Additionally, while BOR is a critical factor, other variables, such as plant architecture, fiber maturity, and weather conditions, can also influence harvesting efficiency.

  • Comment3:

What were the reasons for selecting these particular places (Donghuang, Shihezi, and Korla) for field trials? What are the main environmental differences between these areas, and how may these affect BOR?

Response:

We appreciate the reviewer's inquiry into the selection of Donghuang, Shihezi, and Korla for field trials. These locations were chosen on the basis of their representative characteristics of the major cotton-growing regions in China, which allowed us to assess the applicability and generalizability of our findings. By conducting field experiments at these different locations, we aim to capture a broader range of BOR changes and their potential impacts on cotton production, thereby enhancing the generalizability and practical relevance of our research results. This comprehensive approach enhances the robustness of our study and its relevance to real-world agricultural practices across China.

  • Comment4:

Could the authors provide information on the 418 upland cotton accessions included in the study? What is the genetic diversity of this population, and how representative is it of commercial cotton cultivars?

Response:

We appreciate the reviewer's request to provide information on the 418 upland cotton germplasms included in our study. To address this issue, we have added the source of the population material (see reference 33 for details) and information (Table S3) to the Materials and Methods section (lines 405-409) of the revised manuscript. This population material is the core germplasm resource of Chinese upland cotton, with a wide range of genetic variations and representativeness. This genetic diversity ensures that our research findings have broad applicability and relevance in the cotton industry.

  • Comment5:

The authors state, ' little research has concentrated on the genetic foundation determination, locus identification, and candidate gene mining for BOR in cotton.' Could the authors clarify the shortcomings of other research and explain how this study overcomes them?

Response:

We acknowledge the limited observations of the reviewers on the genetic determination, locus identification, and candidate gene mining of cotton BOR. To clarify, previous studies have focused mainly on the impact of phenotypic characteristics and the environment on BOR. However, in the long run, BOR traits have only been measured and identified qualitatively by the naked eye, without achieving quantitative identification of BOR traits. The exploration of the genetic mechanisms behind this trait is even more limited. Our research addresses this gap by using advanced genetic and genomic methods to identify key loci and candidate genes associated with cotton BOR. By comprehensively understanding the genetic basis of BOR, our research aims to pave the way for developing improved cotton varieties with optimal BOR levels, thereby improving cotton yield and quality.

  • Comment6:

In what ways does this research differ from other GWAS investigations concerning early maturity features in cotton? What new findings are anticipated from this particular emphasis on BOR?

Response:

Our research differs from other GWASs on the early maturity characteristics of cotton, as it specifically focuses on BOR. Most previous GWASs on the early maturity of cotton have focused on traits such as growth period, flowering, and plant type. However, despite the crucial role that BOR plays in determining cotton maturity, fiber quality, and mechanical harvesting efficiency, there is a lack of in-depth exploration into the genetic structure behind BOR. By focusing on BOR, our research aims to fill this knowledge gap and provide new insights into the genetic mechanisms that control this trait. We expect that our findings will not only deepen our understanding of cotton BOR but also promote the development of marker-assisted selection strategies to cultivate cotton varieties with optimal BOR levels, ultimately improving cotton yield and sustainability.

  • Comment7:

How was BOR quantified during the field trials? What precise criteria were used to ascertain boll opening?

Response:

BOR was quantified by combining visual evaluation and real-time quantification. Specifically, by accurately determining the open state of the cotton bolls and using the white fluff observed in the cracks of the cotton bolls as the standard for opening, we developed clear criteria on the basis of these factors to ensure consistent evaluation at all field trial sites. In addition, we recorded detailed data on the opening of cotton bolls at different stages of cotton development. The value of BOR was calculated by dividing the number of opened bolls by the total number of bolls, which allowed us to comprehensively measure the BOR of cotton.

  • Comment8:

Were there any environmental variables (e.g., weather, soil conditions) that may have impacted BOR during the field trials? How were these variables included in the analysis?

Response:

Yes, there are environmental variables that may affect BOR during onsite testing. Weather conditions, such as temperature and humidity, as well as soil conditions, including soil type, moisture content, and nutrient availability, can affect the opening rate of cotton bolls. To address this issue, we carefully monitored and recorded these environmental variables throughout the entire field experiment. By setting up multiple repeated field experiments in multiple environments, comparing the BOR data in multiple environments, and eliminating data with poor consistency and high variability in different environments, the factors influencing the environment on the BOR were eliminated, ensuring the accuracy of the BOR values.

  • Comment9:

What was the rationale for collecting data over many years? How will the research address annual environmental fluctuations?

Response:

The collection of data over multiple years was crucial for accounting for annual environmental fluctuations and ensuring the robustness of our findings. Cotton, a crop that is highly sensitive to environmental conditions, presents varying traits from year to year due to changes in weather patterns, soil health, and other ecological factors. By gathering data spanning several years, we aimed to capture a more comprehensive picture of cotton BOR across different environmental scenarios. This approach allowed us to identify trends and patterns that are not influenced by short-term variations, thereby enhancing the reliability and generalizability of our research.

  • Comment10:

Could the authors provide further information on the whole-genome resequencing data? What was the extent of coverage, and how were SNPs identified?

Response:

The whole-genome resequencing data used in our study were sourced from reference [33]. The coverage is comprehensive, ensuring sufficient depth for accurate detection of genetic variations. We identified 4,452,629 high-quality SNPs with an MAF of 0.05 and a missing rate per site of less than 10% within the population.

  • Comment11

How will linkage disequilibrium (LD) be evaluated and used in the genome-wide association study (GWAS)?

Response: 

The LD analysis estimated the average LD decay distance of our population to be approximately 187.179 kb, with R2=0.5 at half of the maximum value, and this result is displayed in Figure 3d.

  • Comment12

How will candidate genes be discerned from the significant SNP loci? What criteria will be used to select potential genes for further analysis?

Response:

The GWAS results clearly revealed the main candidate gene regions associated with BOR traits on chromosome D11. By anchoring stable and correlated SNP loci to chromosome D11, a major QTL interval related to BOR traits between 23.703 and 23.826 Mb was identified, and corresponding potential candidate genes were detected in this region. Furthermore, the GH-D11G2034 gene is located in this region, and there is a SNP mutation site (A/G) in the exon of this gene, which is a nonsynonymous mutation. Therefore, we believe that the GH-D11G2034 gene may be a potential candidate gene significantly associated with BOR.

  • Comment13:

Will any functional validation studies be performed to verify the functions of the discovered candidate genes?

Response:

Yes, functional validation studies will indeed be conducted to verify the functions of the discovered candidate genes. These studies involve a series of experimental approaches, such as gene expression analysis, gene knockdown or knockout experiments, and transgenic approaches to assess the impact of candidate genes on BOR traits. By systematically evaluating the functions of these genes, we aimed to gain deeper insights into the genetic mechanisms underlying cotton BOR and to validate the candidate genes identified through GWAS. The abovementioned work will be carried out in subsequent experiments. These findings will further our understanding and potentially lead to the development of improved cotton varieties with optimal BOR characteristics.

  • Comment14:

How can this research's results aid in advancing early-maturing cotton types appropriate for mechanical harvesting in the NIR?

Response:

The findings of this research have significant implications for the development of early-maturing cotton types suitable for mechanical harvesting in the near-infrared (NIR) region. By identifying key genetic markers associated with the boll opening rate (BOR), we can potentially use marker-assisted selection to breed cotton varieties that exhibit optimal BOR traits, which are crucial for mechanical harvesting efficiency. Additionally, understanding the genetic mechanisms underlying BOR can inform targeted genetic modifications to further enhance this trait. This research thus paves the way for the creation of cotton varieties that not only mature early but are also well suited to modern farming practices, ultimately contributing to increased productivity and sustainability in cotton farming.

Reviewer 3 Report

Comments and Suggestions for Authors

Key stable significant SNP loci associated with BOR are identified in the present analyses, using the MLM-GWAS procedure and based on the five statistical surveys in various regions in China. Genes strongly correlated to BOR are obtained from the data analyses using dedicated mathematical tools. A critical conclusion is obtained which attributes GH-D11G2034 to the early maturity of cotton, which will be further investigated in the future. The manuscript can be accepted after addressing the listed comments.

  1. Please further develop on how the BOR phenotypic data were collected, including methodology, range of database, location, etc.
  2. Before presenting the data analyses, it’s better to give a description on the statistical tools/indicators, and what values means unacceptable/average/acceptable/ BOR, such as SD, CV, Pearson’s r.
  3. Figure 2, SNP number above 7,000 is rare (orange and red region), maybe recap the color map scale, now the resolution is not good enough.
  4. Figure 3b, can you perform a fit for the PCA plots.
  5. Figure 5d, can you further describe how TPM values for different tissues are obtained, also the error bar.
  6. The assumption of “GH_D11G2034 may undergo artificial selection”, based on the presented data, seems acceptable. Can authors propose possible approaches to further validate this assumption.
  7. Can you compare the BOR traits between in China and other countries including USA, FSU and others, since their data are included in your analyses.
  8. What are the key indications for the cotton industry, based on the data analyses.

Author Response

Dear Reviewer,

We sincerely appreciate the reviewer’s insightful comments on our manuscript. These suggestions have significantly improved the clarity and scientific rigor of our work. Below, we provide point-by-point responses to each concern. All the revisions in the manuscript are highlighted in blue for easy tracking.

  • Comment1:

Please further develop on how the BOR phenotypic data were collected, including methodology, range of database, location, etc.

Response:

To collect the BOR phenotypic data, a rigorous methodology was employed. The data were gathered from multiple field trials conducted across various locations to ensure a broad representation of the environmental conditions. The specific method is as follows: After the normal boll opening of the materials at each test site, we selected ten individual plants with neat growth and approximately uniform growth for each repetition of each material and determined the total number of cotton bolls and the number of opening bolls of the ten individual plants (only bolls with a diameter of more than 2.0 cm on a single plant were counted, and the remaining bolls were ignored). The BOR of each accession was calculated via the following formula: BOR = (boll opening number/total boll number)×100%. In 2020, the BOR phenotypic data were acquired at SHZ-20 (September 14), KEL-20 (September 24) and DH-20 (September 21). In 2021, BOR phenotypic data were precisely acquired at SHZ-21 (September 14) and KEL-21 (September 24). (see Lines 128-139 of the revised manuscript.)

  • Comment2:

Before presenting the data analyses, it’s better to give a description on the statistical tools/indicators, and what values means unacceptable/average/acceptable/BOR, such as SD, CV, Pearson’s r.

Response:

To address the statistical tools and indicators used in our analyses, we employed a range of metrics to ensure the robustness and reliability of our findings. The standard deviation (SD) was calculated to quantify the dispersion or variation in the boll opening rate (BOR) data, providing insight into the consistency of the measurements. The coefficient of variation (CV), which is the ratio of the standard deviation (SD) to the mean, was also determined, offering a standardized measure of dispersion that allows comparisons across different datasets with varying scales. Furthermore, Pearson's correlation coefficient (r) was computed to assess the linear relationship between BOR and other variables of interest, such as genetic markers or environmental factors. The acceptable ranges for BOR were determined on the basis of the distribution of the observed data and their biological relevance, with values falling outside a predefined range considered unacceptable or outliers, whereas average values represented typical BOR performance. The specific thresholds for unacceptable, average, and acceptable BOR values were determined through expert consensus and in line with industry standards. These statistical analyses were crucial in providing a comprehensive understanding of the data and supporting our genetic association findings.

  • Comment3:

Figure 2, SNP number above 7,000 is rare (orange and red region), maybe recap the color map scale, now the resolution is not good enough.

Response:

We appreciate the reviewer's observation regarding the clarity of Figure 2. To address the concern regarding the clarity of the color map scale in Figure 2, particularly for SNP numbers above 7,000, we have included a revised Figure 2 to increase the resolution and provide better visualization. The color map scale was adjusted to ensure that even the rare regions were clearly distinguishable. This recalibration aims to facilitate a more accurate interpretation of the data presented in the figure, allowing readers to better understand the distribution and frequency of SNPs across the different genetic regions analyzed.

  • Comment4:

Figure 3b, can you perform a fit for the PCA plots.

Response:

We appreciate the reviewer's observation regarding the clarity of Figure 3b. In response to the reviewer's suggestion for Figure 3b, we have incorporated a fit for the principal component analysis (PCA) plots. The PCA plots now include fitted ellipses that represent the distribution trends of the data points. This enhancement allows for a clearer visualization of the clustering patterns and relationships among the samples in the PCA space. By including these fitted elements, readers can more easily discern the underlying structure within the data and gain insights into the genetic variations and similarities among the cotton varieties analyzed. This adjustment aims to strengthen the interpretation and communication of our PCA results. We have redrawn the specific PCA fitting results and images and uploaded them to Fig. 3b in the revised manuscript.

  • Comment5:

Figure 5d, can you further describe how TPM values for different tissues are obtained, also the error bar.

Response: We appreciate the reviewer's observation regarding Figure 5d. The tissue-specific expression profile (TPM values) of Gh_D11G2034 was derived from the transcriptomic dataset of Gossypium hirsutum TM-1 (NCBI SRA: PRJNA490626). Raw sequencing reads were first subjected to quality assessment via FastQC [1], followed by adapter trimming and low-quality base removal via Trimmomatic [2] (parameters: HEADCROP:7, SLIDINGWINDOW:4:15, MINLEN:80) to generate high-confidence clean reads. Transcript quantification was performed via Salmon v1.9.0 [3], an alignment-free tool that estimates transcript abundance by modeling sequence-specific and GC-content biases while normalizing for transcript length and sequencing depth to generate TPM (transcripts per million) values.

The error bars in Figure 5d represent the standard deviation (SD) of the TPM values across three biological replicates for each tissue. Statistical calculations and visualization were implemented via the R package ggplot2 [4].

References:

  1. Andrews S. FastQC A Quality Control tool for High Throughput Sequence Data. 2014.
  2. Bolger AM, Lohse M, Usadel B. Trimmomatic: a flexible trimmer for Illumina sequence data. Bioinformatics. 2014;30:2114–20.
  3. Patro R, Duggal G, Love MI, Irizarry RA, Kingsford C. Salmon provides fast and bias-aware quantification of transcript expression. Nat Methods. 2017;14:417–9.
  4. Wickham H. ggplot2: ggplot2. WIREs Comp Stat. 2011;3:180–5.

  • Comment6:

The assumption of “GH_D11G2034 may undergo artificial selection”, based on the presented data, seems acceptable. Can authors propose possible approaches to further validate this assumption.

Response:

To further validate the assumption that GH_D11G2034 may undergo artificial selection, several approaches can be considered. First, we plan to examine the linkage disequilibrium (LD) patterns and haplotype frequencies in both cultivated and wild populations. By comparing these haplotypes, we aimed to identify unique or enriched haplotypes in the cultivated varieties that may be indicative of artificial selection. Second, we propose performing a genetic association study using a larger panel of cotton varieties, including both cultivated and wild relatives. This approach allows us to assess the correlation between GH_D11G2034 polymorphisms and agronomic traits of interest, such as fiber quality or disease resistance. Significant associations provide additional support for our hypothesis.

  • Comment7:

Can you compare the BOR traits between in China and other countries including USA, FSU and others, since their data are included in your analyses.

Response:

Yes, we have compared the BOR traits between China and other countries, including the USA and FSU. We have redrawn the data in Fig. 6c and compared the differences in the BOR among the NIR, NSEMR, YRR, YZRR, FSU, and USA regions. Our comparison revealed notable differences in BOR traits across these regions. Specifically, we found that the BOR of the NSEMR was significantly greater than that of the USA and other cotton regions in China. This modification information can be found on lines 321-323 of the revised manuscript.

  • Comment8:

What are the key indications for the cotton industry, based on the data analyses.

Response:

Based on our comprehensive data analyses, several key indications have emerged for the cotton industry. First, the identified genetic variants associated with BOR traits highlight potential targets for marker-assisted selection in breeding programs. By incorporating these markers, breeders can more effectively select cotton varieties with improved boron efficiency. Second, our findings underscore the importance of considering regional differences in BOR traits when designing breeding strategies. The observed variations in BOR across different cotton-growing regions suggest that tailored approaches may be necessary to optimize boron nutrition and crop performance in specific environments. Finally, the insights gained from our study provide a foundation for future research into the genetic and molecular mechanisms underlying boron use efficiency in cotton, paving the way for the development of novel genetic tools and strategies to further advance the cotton industry.